Automatic cassava disease recognition using object segmentation and progressive learning

Che Chang 1 2
http://orcid.org/0000-0001-6108-2562 Xue Nian 3 xuenian@sdut.edu.cn
Li Zhen 3 legion@sdut.edu.cn
Zhao Yilin 4
Huang Xin 5
1 Electronic and Information Engineering, School of Civil Engineering, Harbin University , Harbin, Heilongjiang , China
2 Heilongjiang Urban Water Quality Monitoring Co, Ltd , Harbin, Heilongjiang , China
3 School of Computer Science and Technology, Shandong University of Technology , Zibo, Shandong , China
4 Department of Computer Science and Engineering, Tandon School of Engineering, New York University , New York, Brooklyn , United States
5 Taiyuan University of Technology , Taiyuan, Shanxi , China
Angiulli Giovanni
Electronic publication date: 2025 Mar 18
Publication date: 2025
Volume: 11
Electronic Location ID: e2721
Received 2024 Nov 20; Accepted 2025 Jan 30
Copyright: © 2025 Che et al.
Copyright year: 2025
Copyright holder: Che et al.
License: This is an open access article distributed under the terms of the Creative Commons Attribution License, which permits unrestricted use, distribution, reproduction and adaptation in any medium and for any purpose provided that it is properly attributed. For attribution, the original author(s), title, publication source (PeerJ Computer Science) and either DOI or URL of the article must be cited.
License URL: https://creativecommons.org/licenses/by/4.0/

Keywords: Cassava disease recognition, Progressive learning, Object segmentation, Deep learning, Computation efficiency

Funding: Key Topics for the 14th Five Year Plan of Education Science in Heilongjiang Province in 2022 Research on the Path of Building a Smart Classroom Teaching Model and Gold Course by Drawing on the Education Experience of FH in Germany under the Background of Professional Certification GJB1422322 This research work is supported by the Key Topics for the 14th Five Year Plan of Education Science in Heilongjiang Province in 2022: Research on the Path of Building a Smart Classroom Teaching Model and Gold Course by Drawing on the Education Experience of FH in Germany under the Background of Professional Certification (Subject No: GJB1422322). The funders had no role in study design, data collection and analysis, decision to publish, or preparation of the manuscript.

==============================
Cassava is a vital crop for millions of farmers worldwide, but its cultivation is threatened by various destructive diseases. Current detection methods for cassava diseases are costly, time-consuming, and often limited to controlled environments, making them unsuitable for large-scale agricultural use. This study aims to develop a deep learning framework that enables early, accurate, and efficient detection of cassava diseases in real-world conditions. We propose a self-supervised object segmentation technique, combined with a progressive learning algorithm (PLA) that incorporates both triplet loss and classification loss to learn robust feature embeddings. Our approach achieves superior performance on the Cassava Leaf Disease Classification (CLDC) dataset from the Kaggle competition, with an accuracy of 91.43%, outperforming all other participants. The proposed method offers a practical and efficient solution for cassava disease detection, demonstrating the potential for large-scale, real-world application in agriculture.

Introduction

Cassava is a vital crop for food security, serving as a primary dietary source for over half a billion people worldwide, particularly in sub-Saharan Africa, where it is the second-largest contributor to carbohydrate intake after maize (Dixon & Ssemakula, 2008). As a staple crop, cassava is cultivated by smallholder farmers, serving as a vital component of food security. Its ability to thrive under harsh conditions such as drought and poor soil quality makes it an essential crop for smallholder farmers, many of whom depend on cassava cultivation for their livelihoods. Notably, this starchy root crop is cultivated on approximately 80% of household farms in Sub-Saharan Africa (Spencer & Leone, 2005). However, the widespread prevalence of cassava diseases poses a significant threat, leading to a gradual decline in cassava production and causing substantial economic harm to farmers and related industries. Hence, a prompt, accurate, and reliable diagnosis of cassava diseases is crucial to implement effective mitigation measures and minimize losses. Despite its resilience, cassava is highly susceptible to devastating diseases, most notably cassava mosaic disease (CMD) and cassava brown streak disease (CBSD), which have severe impacts on yields and pose significant threats to food security and local economies (Sidik, 2020).

Conventional diagnostic methods for cassava diseases can be broadly categorized into several approaches. These include in-field diagnosis, grafting diagnostics, serological diagnosis, electron microscopy observation (Bock et al., 2010; Legg & Thresh, 2003). These techniques collectively provide a range of options for identifying and characterizing cassava diseases. Among the various diagnostic methods, the in-field diagnostic approach stands out as the most commonly utilized method for cassava diseases. This technique enables the differentiation of four main symptoms: cassava bacterial blight (CBB), cassava green mottle (CGM), CBSD and CMD (Sidik, 2020). However, in-field diagnosis is associated with certain limitations. These include low detection precision, high labor costs, and a lengthy and complex detection process that often requires specific laboratory conditions (Rwegasira, Rey & Nawabu, 2011). These challenges arise due to the reliance on naked-eye observation, which is subjective and influenced by individual experiences and knowledge. Moreover, the irregular patterns and variations in disease phenotypes observed in cassava leaves and roots further complicate the process (Sidik, 2020).

Advanced molecular nucleic acid-based methods such as DNA probe hybridization, polymerase chain reaction (PCR) amplification, and nucleotide sequencing involve intricate procedures and higher detection costs, such as the need for specialized equipment and extended time periods (Legg & Thresh, 2003; Thresh & Fargette, 2003; Bové et al., 1993). They provide greater accuracy but are not feasible for widespread field application due to their complexity and cost. Consequently, none of the existing diagnostic methods have been efficiently adopted in cassava production. Furthermore, there is currently no standardized, accurate, and cost-effective method available for reliable in-field diagnosis of cassava diseases (Legg et al., 2015; Rwegasira, Rey & Nawabu, 2011).

Consequently, there is an urgent need for an accurate, reliable, and cost-effective method for diagnosing cassava diseases in-field, especially for resource-constrained farmers. As of now, the available technologies for detecting cassava diseases rely heavily on the assistance of government-funded agricultural experts who visually inspect and diagnose the plants (Rwegasira, Rey & Nawabu, 2011). However, this approach is characterized by limited availability, high labor requirements, and significant expenses. Moreover, practical solutions for farmers must be designed to perform well under challenging conditions, as African farmers may only have access to low-resolution, low-bandwidth mobile-quality cameras.

In this study, we propose a novel deep learning-based approach that leverages efficient networks to reduce computational costs while maintaining high performance in image recognition tasks. Our system aims to classify cassava leaf images into four disease categories and a healthy class, enabling farmers to make prompt decisions and mitigate damage. By integrating advanced computer vision techniques, our approach offers a practical, scalable solution for cassava disease detection in the field.

Our article makes several significant contributions, which are outlined below: Application of computer vision techniques in real-world cassava disease recognition: There are limited reports on the use of computer vision techniques for cassava disease recognition in practical settings. Our experimental results demonstrate that our proposed computerized cassava disease recognition method achieves more than 90% accuracy within a response time of 1 s. This outperforms traditional, time-consuming methods that typically yield accuracies below 75%.

Introduction of a novel self-supervised segmentation algorithm: We propose a new self-supervised segmentation algorithm and methodology that effectively combines classification loss and triplet loss to improve the learning of feature embeddings. This approach enhances the overall recognition accuracy by 3%.

Design of a deep learning ensemble with efficient networks: We develop a deep learning ensemble based on efficient networks, specifically the EfficientNet series (Tan & Le, 2019). Our ensemble achieves a high level of efficiency for real-world applications by striking a balance between using lightweight models and maintaining strong overall performance.

The remaining sections of this article are structured as follows: ‘Related Work’ reviews related work, ‘Materials and Methods’ describes the proposed methodology, ‘Results and Discussions’ presents experimental results, and the ‘Conclusion’ concludes the article with discussions on future work.

Related work

The use of machine learning (ML) and deep learning (DL) techniques in plant disease detection has seen a significant rise, with the aim of automating and improving accuracy in agricultural applications. In the case of cassava disease diagnosis, researchers have explored both traditional ML techniques and more recent DL-based approaches, each with their advantages and limitations.

Early works, such as that by Roslan et al. (2021), applied support vector machines (SVM) for cassava disease diagnosis by manually extracting features from leaf images. This method achieved a respectable accuracy of 87.5%, but it was computationally expensive and required significant human effort. Similarly, Ghazalli & Roslan (2022) used SVMs to detect bacterial diseases in cassava plants, achieving high accuracy but facing similar challenges of high computational costs and manual feature extraction.

Recent advancements in machine learning (ML) and deep learning (DL) technologies have opened up new possibilities for automatic plant disease detection. Convolutional Neural Networks (CNNs), in particular, have demonstrated impressive performance in detecting plant diseases from leaf images (Mohanty, Hughes & Salathé, 2016). However, applying these technologies to cassava disease recognition remains a relatively unexplored area. Previous studies have shown the technical feasibility of using automated image recognition systems for disease detection in crops like citrus and cassava (Deng, Li & Hong, 2014; Sidik, 2020), but challenges remain in deploying these solutions in real-world, resource-limited environments, especially those with variable lighting and low-resolution camera capabilities (Ramcharan et al., 2017). Sidik (2020) proposed a Bayesian expert system for diagnosing cassava diseases, which continuously updates its knowledge base with new data from the Internet. While this method can improve diagnostic accuracy over time, it may not be suitable for resource-limited environments or scenarios requiring real-time diagnosis, such as with smartphone applications. Similarly, Ramcharan et al. (2019) applied a CNN-based object detection model in a mobile app for cassava disease detection in Tanzania. While promising in controlled environments, the performance of the model dropped significantly under real-world field conditions. Bock et al. (2010) explored hyperspectral imaging for plant disease detection, which can provide more accurate results than RGB imaging. However, the high cost of hyperspectral imaging equipment limits its practical use in large-scale, low-resource agricultural settings. Other studies, like those by Deng, Li & Hong (2014), have focused on diseases in other crops but provide valuable insights for generalizing DL models across plant species.

In the field of computer vision, diagnosing cassava diseases primarily involves object recognition through image analysis. Unlike conventional object recognition approaches (Zhang et al., 2009; Li, Crandall & Huttenlocher, 2009), which focus on improving recognition performance through carefully designed statistical models, cassava disease recognition requires a fast response time coupled with high recognition accuracy. To address this challenge, a deep learning ensemble utilizing efficient networks is proposed in this study to reduce computational costs while enhancing overall performance in image recognition tasks. The objective of this study is to classify each cassava image into one of four disease categories, with a fifth category indicating a healthy leaf. By leveraging the proposed machine learning-based approach, farmers would be able to promptly identify cassava diseases, potentially mitigating irreversible damage to their plants. The integration of efficient networks and deep learning techniques holds promise for enabling rapid and accurate disease identification, providing farmers with timely information to take appropriate actions and safeguard their crops.

The field of camouflaged object detection (COD) shares significant parallels with cassava leaf disease detection, particularly in addressing the challenges posed by complex, cluttered, or low-contrast backgrounds. Several cutting-edge strategies from COD research offer insights that could inform the development of enhanced methodologies for our task. Techniques such as wavelet decomposition, as introduced in Jiang et al. (2023), He et al. (2023b), emphasize the importance of separating high-frequency details (e.g., edges and textures) from low-frequency background information. This decomposition helps isolate critical features for object detection while suppressing irrelevant noise. Such methods could be beneficial in cassava disease detection, where identifying subtle leaf pattern variations is critical. Multi-scale feature grouping, as detailed in Ma et al. (2020), He et al. (2024), addresses the challenge of capturing features across different resolutions. By aggregating multi-scale features, this approach enhances the detection of concealed objects against varied backgrounds. For cassava leaves, this strategy could improve robustness against occlusions, lighting variations, and background clutter by leveraging features from multiple spatial scales. Incorporating auxiliary tasks has shown promise in COD to enhance primary objectives. For example, the auxiliary edge reconstruction task described in He et al. (2023c), Liu et al. (2024) demonstrates the effectiveness of training the model to reconstruct object boundaries. Applying a similar approach in cassava leaf disease detection could help delineate leaf edges from complex or noisy backgrounds, improving the model’s ability to focus on disease-specific features. These advanced techniques underscore the potential for integrating principles from COD research into agricultural applications. While not implemented in the current work, their relevance highlights promising directions for future research and model enhancements.

Enlightened by these works, our study proposes a novel self-supervised segmentation approach combined with EfficientNet (Tan & Le, 2019), which balances accuracy and computational efficiency, making it more suitable for mobile deployment in resource-limited environments. The proposed method addresses key issues such as environmental variability and limited computational power, offering a more robust solution for real-world cassava disease diagnosis.

Materials and Methods

One of the key contributions of our article is the development of the first deep learning approach for the recognition of common cassava leaf diseases. Our approach offers both fast and accurate disease recognition. Specifically, we extracted the foreground cassava leaf object out of the cluttered in-field image based on Image Segmentation with Automatic Weight Optimization (GMM), and designed a new self-supervised segmentation approach based on U-Net, followed by the ensemble based on deep learning models to achieve the cassava disease classification.

Datasets and protocols

For our study, we utilize an official cassava dataset consisting of 21,367 labeled images, which was specifically curated for the Kaggle competition (https://www.kaggle.com/c/cassava-leaf-disease-classification/data) (Mwebaze et al., 2020). The dataset was gathered through a regular survey conducted in Uganda, Africa. The majority of the cassava images in the dataset were crowdsourced from cassava producers who captured images from their own gardens. These images were then annotated by experts at the National Crops Resources Research Institute (NaCRRI) in collaboration with the Makerere Artificial Intelligence (AI) Lab at Makerere University in Kampala, Uganda. This dataset provides a comprehensive and diverse collection of cassava images, enabling us to train and evaluate our cassava disease recognition system effectively. By utilizing this dataset, we aim to contribute to the advancement of cassava disease recognition technology and facilitate practical applications in the agricultural domain.

The dataset used in our study consists of five categories: Cassava Bacterial Blight (CBB), Cassava Green Mottle (CGM), Cassava Brown Streak Disease (CBSD), Cassava Mosaic Disease (CMD), and healthy leaves without any disease. These categories were assigned based on the judgments of agricultural experts. Figure 1 illustrates some representative images from each of these five categories. The five categories are defined based on their health status and distinctive symptoms. CBB is marked by angular, water-soaked spots that turn brown with yellow borders, accompanied by yellowing leaves, wilting, and branch dieback. CGM affects young leaves with puckering, yellow patterns, mosaic-like green patches, twisted margins, and occasional severe stunting. CBSD exhibits yellow or necrotic vein banding that may coalesce into large yellow patches, without leaf distortion. CMD is distinguished by severe mosaic symptoms, including mottling, discoloration, and malformation of leaves. Healthy leaves lack these pathological features and mostly appear uniform and undistorted.

Figure 1 Sample images of the CLDC Leaf-5 dataset.

A summary of the dataset statistics is presented in Table 1. Notably, the dataset exhibits a significant class imbalance. Only approximately 12% of the dataset comprises images of healthy leaves, while the majority of the images represent diseased leaves. Class 3, in particular, contains more samples than all the other classes combined, thereby adding to the complexity of the classification task. The presence of class imbalance poses a challenge in training and evaluating our cassava disease recognition system. However, by addressing this issue and leveraging advanced machine learning techniques, we aim to develop an effective solution that can accurately identify different types of cassava diseases in real-world scenarios.

Table 1 Statistics for the cassava leaf dataset.

Classes	#no.	%	
0: Cassava bacterial blight (CBB)	1,087	5.1%	
1: Cassava brown streak disease (CBSD)	2,189	10.2%	
2: Cassava green mottle (CGM)	2,386	11.2%	
3: Cassava mosaic disease (CMD)	13,158	61.5%	
4: Healthy	2,577	12%	

To simulate the automatic monitoring of cassava leaves in real-life field conditions, volunteers were instructed to capture photos under various conditions, including different weather conditions, illuminations, resolutions, and viewpoints. The challenging conditions in the dataset training images primarily encompass low lighting, which may affect the image either partially or globally; noise, often manifesting as blurriness; and cluttered backgrounds, which may include unrelated objects, large non-relevant areas, human presence, or shadows causing occlusions. Representative examples of these adverse conditions are illustrated in Fig. 2.

Figure 2 Examplar images of adverse conditions.

Cassava leaf disease recognition approach

In our approach to cassava leaf disease recognition, we first isolate the cassava leaf from complex backgrounds to improve classification accuracy, which would be compromised if performed on the entire image. This isolation begins with traditional clustering techniques, using Gaussian mixture models (GMM), to segment leaf objects, as is an effective pre-processing technique. However, limitations in speed and precision arise with GMM alone. To address these, we incorporate a self-supervised U-Net model, fine-tuned for efficient and precise segmentation. Once segmented, the cassava leaf images undergo feature embedding through a progressive learning algorithm (PLA) to generate a high-dimensional feature vector optimized for disease classification. Finally, we use an ensemble of pre-trained deep learning models to classify the segmented leaf images, achieving a robust framework for accurate and scalable disease detection.

Preprocessing by leaf object segmentation

Before the cassava disease recognition step, it is of vital importance to carry out localization of the cassava leaf object from the cluttered background. Otherwise, if the classifier is applied directly to the whole image, the recognition accuracy of the system will be drastically reduced due to the cluttered background. In the applications of plants, methods have been developed to highlight green vegetation in imagery from widely-used RGB cameras. It has been demonstrated that plant health can be evaluated via the Normalized Difference Vegetation Index (NDVI) (Pettorelli, 2013).

In this work, by applying GMM (Deng, Li & Hong, 2014) based on image feature vectors, we can separate the foreground, i.e. the cassava leaf part, from the cluttered background by solely using traditional image processing methods. Color feature is obtained by the Elliptical Color Index (ECI) (Lee, Golzarian & Kim, 2021) and Color Index Vegetation Extraction (CIVE) (Kataoka et al., 2003), and texture feature is obtained by Gabor filter (Bovik, 1991), respectively. A texture-based feature vector which is 24-dimensional is obtained for each image pixel using a convolution based on Gabor filtering. Given R, G, and B channels of an image, first Gaussian filtering is applied to blur an image, and then vegetation index is calculated, followed by normalization to the range of [0,255]. The ECI and CIVE color indices are defined as follows:

(1) ECI=(R−1)2+G2/0.16

(2) CIVE=0.441∗R−0.881∗G+0.385∗B+18.787.

To brief the GMM algorithm, the first two sets of segmented regions are obtained which are partitioned by clustering according to the color feature (2-dimension consisting of ECI and CIVE) and the texture feature (24-dimension), respectively. By iteratively maximizing the similarity between the two sets of segmented regions according to the similarity measure, we can obtain the optimized weight vector. Then the color and texture features are combined with the obtained optimal weights and perform the final clustering based on the combined feature vector. We denote V={v1,v2,⋯,vN} as the set of color or texture feature vectors, then the probability density of a GMM containing K components can be formulated as

(3) p(v)=∑j=1kπjϕΣj(v−cj)

(4) ϕΣj(v−cj)=12dπd|Σj|exp⁡(−12(v−cj)TΣj−1(v−cj))

where πj is a mixing coefficient of one of the different Gaussian filtering components, which are positive and sum to one, cj is the centroid vector of each component, Σj denotes a covariance matrix, and ϕΣj(v−cj) denotes multivariate normal density.

Regarding the estimation of the parameters πj, vj and Σj, the expectation maximization (EM) technique is utilized for maximizing the log-likelihood. The EM procedure is carried out iteratively after initialization. Firstly, EM performs E-step, which gives an estimation of the color vector probability density vi of the component cj of the t-th iteration:

(5) p(t)(cj∣vi)=p(vi∣cj)/∑l=1Kp(vi∣cl)

(6) sj(t)=∑i=1Np(vi∣cj)/∑l=1Kp(vi∣cl)

Secondly, the parameters of GMM are then updated during the M-step as follows:

(7) πj(t+1)=1Nsj(t)

(8) cj(t+1)=1sj(t)∑i=1Nvi⋅p(t)(cj∣vi)

(9) Σj(t+1)=1sj(t)∑i=1Np(t)(cj∣vi)(vi−cj(t+1))(vi−cj(t+1))T

In this work, as the cassava image content can be roughly divided into foreground/background parts, we set K=2 as the number of components. Once the cassava image is converted to the two-component GMM representation, we select the image pixels that correspond to the component with higher values in the green channel to be the foreground cassava leaf. Figure 3 depicts the process of the foreground detection, where the original and the extracted cassava images are shown in the left and the right, respectively.

Figure 3 Foreground object extraction from cassava images using GMM segmentation.

Leaf object segmentation by deep learning

Now the foreground cassava leaf object is obtained by GMM. However, there are some disadvantages: (1) There are some artifacts in object segmentation, e.g., the leaf objects have small holes identified as background object parts. This is due to the lack of noise reduction in traditional object segmentation approaches, and (2) The GMM process is rather slow although it is visually accurate. This is because of the iterative optimization process using traditional feature extraction.

In view of these issues, we propose a novel self-supervised object segmentation approach. Firstly, we apply GMM to infer the pseudo-truth segmentation masks of the cassava images. This is done by randomly cropping 512×512 images from the original images, before extracting features and applying GMM segmentation to generate the vegetation mask. As ECI and CIVE only work correctly on healthy plants, all diseased samples are filtered out, but data augmentation techniques are applied to simulate diseases. Secondly, the vegetation mask of each 512×512 image sample will be kept as a ground truth mask, and fed into a U-Net (Ronneberger, Fischer & Brox, 2015) for training the segmentation model via deep learning.

We specially designed an enhanced U-Net structure which shows visually accurate segmentation with reasonable computation efficiency, which is shown in Fig. 4. The enhanced U-net is essentially comprised of four different convolution modules: (1) Base module; (2) dense module; (3) transition down module; and (4) transition up module, as shown in Fig. 5. The enhanced U-Net has deeper convolutions and seven intermediate loss values during the transition down and transition up stages are summed up to the final stage loss value, enabling the U-Net to achieve more stable training and better accuracy.

Figure 4 Network structure of the proposed U-Net.

Figure 5 Modules of the proposed U-Net: (A) base module; (B) transition up module; (C) dense module; and (D) transition down module.

In order to reduce the object boundary artifact for better classification, we masked the background with pure green color. Typical segmentation results from the proposed U-Net are illustrated in Fig. 6. The original images, GMM segmented images, and U-Net segmented images are juxtaposed in Fig. 7.

Figure 6 Object-segmented images based on the proposed self-supervised U-Net.

Figure 7 Foreground object extraction from a cassava leaf image.

Learning feature embedding

For images masked by foreground cassava leaf objects, the progressive learning algorithm (PLA) (Li et al., 2021, 2022) is exploited to learn the discriminative capability. Figure 8 illustrates the architecture, where the final feature vector is 1,024-dim that is produced by 1×1 convolution of the feature vectors followed by global pooling. These features are optimized through a combination of cross-entropy loss and triplet loss in the back-propagation of the training process. The final 1,024-dim feature embeddings will be generated by feature concatenation, and used for inference.

Figure 8 Deep ensemble network.

Triplet loss distinguishes itself from traditional classification loss by producing the feature embedding vectors (Schroff, Kalenichenko & Philbin, 2015; Shi et al., 2016; Hermans, Beyer & Leibe, 2017), and the metric embedding mapping between the original image data and the optimal semantic features is learned: g(θ):RD→RF, where RD represents the manifold of the image data, RF represents the space of semantic features, and x is the input data. We define θ as the network parameters. In this work, the non-linear function g(θ) is implemented through a deep learning network. The metric mapping is defined as Di,j=D(gθ(u),gθ(v)):RD×RD→R.

The concept of triplet loss originates from the Large Margin Nearest Neighbor (LMNN) loss (Weinberger, Blitzer & Saul, 2005):

(10) LLMNN(θ;X)=(1−μ)∑ya=ybDa,b+μ∑a,b,nya=yb≠yn[m+Da,b−Da,n]

where a and b are samples from an identical class, while m and μ represent the margin parameter and the weighting factor, respectively. Ideally, based on such loss functions, g(θ) is likely to cluster images that are semantically similar onto adjacent feature points in the metric space, and disperse semantically dissimilar images onto distant points in the metric space.

In PLA (Li et al., 2021, 2022), two progressive parameters p and k are introduced to control the process when optimizing the triplet loss. Let k∈[1,…,K], p∈[1,…,P], and Largestk(⋅) and Smallestp(⋅) represent the value picking operations upon the batch Da,b which ranks the k-th largest, and the value from Da,n which ranks the p-th smallest, respectively. Consequently, the original triplet loss is generalized as

(11) LGBHk,p(θ;X)=∑l=1P∑a,bya=yb=lln(1+em+Tk,p(a,b,n))

(12) Tk,p(a,b,n)=Largestk(Da,b)−Smallestpyn≠ya⁡(Da,n).

Any discrete hard level can be configured by adjusting the parameters k and p of the above generalized batch hard loss. When k=1, only the hardest examples, which might even be outliers, are used for training, resulting in an unstable training process as in batch hard loss. However, when k>1, the too-hard or wrong labels are circumvented, and the unstable training situation is alleviated.

Furthermore, the classification loss is integrated to smooth the loss and ameliorate side-effects of the over-difficult training samples. The classification loss is normalized via Softmax:

(13) Lsoftmax(θ;X)=−∑i=1PKlneWyiTfi+byi∑j=1MeWyjTfj+byj

where M denotes the total classes of the training dataset, and W represents the weights of a 1×1 convolution function.

Consequently, the composite objective function becomes the combination of the cross-entropy classification loss and the generalized triplet loss as follows:

(14) Lk,p(θ;X)=Lsoftmax(θ;X)+λLGBHk,p(θ;X)

where λ denotes a predefined trade-off factor to balance the generalized triplet loss and the conventional classification loss. It is determined by a Bayesian optimization procedure in Li et al. (2021).

The whole training procedure is carried out in an end-to-end manner as shown in Algorithm 1. Considering a minimal requirement of memory usage, we set P=4, K=8 to as the training image batches parameters.

Algorithm 1 Cassava classifier training algorithm.

Input: From the training set, construct mini-batches containing P=4 randomly chosen classes and K=8 randomly chosen images per class.	
Output: The optimal CNN weights that will produce 1,024-dim feature embedding.	
Initialization: Initialize N sets of random hyper-parameters W={w1,w2,⋯,wN} where wi=(λi,mi,ki,pi), λi∈[0,2], mi∈[−0.1,0.3],	
ki∈[1,8], pi∈[1,4], for i=1,⋯,N.	
Repeatably Do	
 for i=1 to N:	
  Exploration: Back-propagate network for 10 epochs and evaluate L via Eqs. (11) and (14), followed by the evaluation of Bayesian objective f(wi).	
  Restoration: network weights are recovered to the values prior to the 10 exploration epochs.	
 end for	
  Exploitation: Generate a new improved version of candidate w′ according to f(W), followed by the Gaussian update process, and add w′ to W.	
  Perform back-propagation to enable network weights updating through 300 epochs once the hyperparameter w^ is updated;	
  Record the model weights of the lowest L corresponding to the current hyperparameter w^;	
Until the maximum number of epochs ( M=1,000) is finished.	

Deep learning ensemble

After the PLA model is trained, the inference of each object masked cassava image recognition will generate a 1,024-dim feature. There are a lot of choices of backbones of the deep learning network. Based on preliminary experiments, we have selected six state-of-the-art networks: ResNet-50 (He et al., 2016), and EfficientNet series (b3, b4, b5, b6, b7) (Tan & Le, 2019).

The choice of ResNet-50 and EfficientNet as the backbone architectures for our proposed algorithm is guided by their proven effectiveness in deep learning tasks, particularly in image classification and feature extraction.

ResNet-50’s architecture, with its innovative use of skip connections, effectively addresses the vanishing gradient problem, enabling the training of deep networks without degradation in performance (He et al., 2016). This is particularly critical for the cassava disease detection task, where the model must discern subtle variations in leaf patterns and textures. ResNet-50 is widely recognized for its robust performance in image classification tasks and has been extensively applied in various industries, including medical imaging (Fu et al., 2020), autonomous vehicles (Grigorescu et al., 2020), and facial recognition systems (Wang et al., 2018). Its architecture, featuring 48 convolutional layers, is designed to extract rich and hierarchical features from input images while addressing the vanishing gradient problem through skip connections. Studies have shown that ResNet-50 is effective for specialized applications such as fire detection and remote sensing (Jabnouni et al., 2022).

EfficientNet, on the other hand, employs a compound scaling strategy that uniformly scales network depth, width, and resolution. This approach enables EfficientNet to achieve state-of-the-art performance with significantly fewer parameters and FLOPs (floating point operations per second) compared to ResNet-50. For example, EfficientNet-B0 achieves comparable accuracy to ResNet-50 with only 5.3 million parameters vs ResNet-50’s 25.5 million, making it computationally more efficient (He et al., 2016; Tan & Le, 2019). This characteristic is valuable for handling the inherent challenges of cassava disease detection, such as the variability in image quality due to diverse lighting conditions and occlusions. EfficientNet represents an advancement over ResNet-50 by introducing a compound scaling method that uniformly scales network depth, width, and resolution, achieving state-of-the-art performance with fewer parameters and computational resources (Tan & Le, 2019). By leveraging EfficientNet, we ensure that the model is both accurate and computationally efficient, aligning with the practical constraints of real-world deployment in agricultural settings. These architectures have been extensively validated in similar classification tasks, further supporting their suitability for this study.

The demonstrated effectiveness and efficiency of ResNet-50 and EfficientNet architectures, supported by their extensive validation in similar classification tasks, justify their adoption for the cassava leaf disease detection task in this study.

Since pre-trained models often exhibit good performance for general image classification tasks (Geng et al., 2016; Zheng, Zheng & Yang, 2016), while niche approaches use networks trained from scratch (Li et al., 2014; Ahmed, Jones & Marks, 2015; Cheng et al., 2016; Xiao et al., 2016; Shi et al., 2016; Varior, Haloi & Wang, 2016), we prefer the pre-trained model of each selected network. We train the backbone network from a pre-trained ImageNet model, with the classifier layer of 1×1 convolutions randomly initialized. The 1,024-dim feature embeddings are obtained from each of the six models.

Results and discussions

In this section, we provide implementation details and evaluation metrics, followed by a comprehensive performance analysis of our proposed method and compare it with other state-of-the-art methods in the field.

Implementation details

The preliminary experiments were conducted on a system equipped with an Intel Core i7-9700K CPU running at 3.6 GHz, 16 GB of RAM, and an NVIDIA GTX 2080Ti GPU with 11 GB of VRAM. The software environment consisted of TensorFlow 2.4 and Python 3.8, running on Ubuntu 20.04. The final score in Kaggle competition was obtained using Kaggle free computing environment of NVIDIA TESLA P100 GPUs. The two environments have consistent results.

All images are resized according to network input shapes, which may vary in different networks with regard to trade-off between efficiency and accuracy. We first resized the image to target shapes, then did a series of image augmentation steps in the training process.

The training dataset, consisting of 21,367 images, is split in a stratified manner into 90% for training and 10% for validation. The testing dataset comprises 15,000 images, which is set by the Kaggle platform. During the training process, five-fold stratified cross-validation is employed to ensure robust evaluation.

The backbone models, ResNet-50 (He et al., 2016) and the EfficientNet series (Tan & Le, 2019), are implemented following the methodologies outlined in their respective original articles. Training details, including hyperparameter settings, model architectures, and training protocols, remain consistent with the original implementations. For further information, please refer to the original publications.

We use Adam (Kingma & Ba, 2014) as the optimizer to learn feature embedding and set β1=0.9 within 150 epochs and β1=0.5 for remaining epochs, and β2=0.999. We followed the learning rate scheduler as proposed by Hermans, Beyer & Leibe (2017):

(15) α(e)={α0ife≤e0α0×0.001e−e0e1−e0ife0≤e≤e1

where we set α0=3×10−4, e0=150 epochs, and e1=300 epochs.

The image resolutions and classification loss functions for different base networks selected for the ensemble are listed in Table 2. The classification loss has two options: cross entropy loss (Li et al., 2020) and bi-tempered (BiT) logistic loss (Amid et al., 2019). For further details, refer to the implementation code in the Kaggle competition platform (https://doi.org/10.5281/zenodo.14739855).

Table 2 Configuration of different base networks.

Base networks	Image size	Classification loss	
ResNet-50 (He et al., 2016)	(512, 512)	CrossEntropy Loss	
EfficientNet-b3 (Tan & Le, 2019)	(300, 300)	BiT (t1 = 0.8, t2 = 1.4)	
EfficientNet-b4 (Tan & Le, 2019)	(512, 512)	BiT (t1 = 0.8, t2 = 1.4)	
EfficientNet-b5 (Tan & Le, 2019)	(468, 468)	BiT (t1 = 0.8, t2 = 1.4)	
EfficientNet-b6 (Tan & Le, 2019)	(528, 528)	BiT (t1 = 0.8, t2 = 1.4)	
EfficientNet-b7 (Tan & Le, 2019)	(600, 600)	BiT (t1 = 0.8, t2 = 1.4)	

Data augmentation

Although challenging conditions such as low light, noise, and cluttered backgrounds are already present in the training image dataset, the model trained on this dataset demonstrates resistance to these conditions to some extent.

To further enhance its robustness, we incorporate a range of image augmentation techniques during the training process. These augmentations include random cropping, random flipping, hue saturation value (HSV) adjustment, random brightness-contrast adjustment, normalization (with mean = [0.485, 0.456, 0.406] and standard deviation = [0.229, 0.224, 0.225]), and dropout. Random erasing (Zhong et al., 2017) is also employed, which randomly ( p=0.5) masks portions of cassava leaves in the images. These data augmentation strategies aim to simulate a wide variety of real-world conditions, enhancing the model’s ability to focus on disease-relevant features while ignoring extraneous factors. By diversifying the training data, these techniques enhance the model’s generalization capabilities, enabling it to perform reliably across diverse environmental conditions and image quality scenarios.

Specifically, these data augmentation techniques applied during the training process aim to enhance the model’s generalization capabilities: (1) HSV adjustment, Random Brightness-Contrast adjustment: These augmentations simulate lighting variations, including low light conditions and overexposure, to improve the model’s robustness under diverse illumination scenarios. Brightness adjustments also introduces variability in image brightness, mimicking noise caused by lighting inconsistencies. (2) Random erasing: This technique masks parts of cassava leaves, emulating occlusion and missing data, which indirectly aids in generalizing to noise-like conditions and helps the model generalize by masking parts of the background. (3) Normalization: Normalizing image intensities to a standard mean and standard deviation reduces sensitivity to global lighting differences.

Furthermore, integrating restoration techniques, such as illumination restoration based on latent diffusion model (LDM) (He et al., 2023a), could enhance preprocessing steps and improve the model’s performance under extreme degradations. Since maintaining short inference times for efficiency is critical for real-world deployment, this preprocessing step in inference phase is skipped. However, in cases of severe low-light conditions, the diffusion-based image restoration such as He et al. (2023a) could be integrated into the system as a preprocessing step to improve robustness.

Although there is a significantly higher representation of Class 3, the dataset’s imbalance also reflects the natural distribution of cassava leaf disease classes observed in real-world scenarios. This alignment suggests that the private test set is likely to follow a similar class proportion, minimizing concerns about dataset imbalance in practical applications. Nevertheless, steps were taken to address the imbalance in the training dataset to ensure optimal model performance.

To address the class imbalance, additional data augmentation techniques were applied to the underrepresented classes. The augmentation aimed to increase the diversity and quantity of data for these classes, reducing the likelihood of the model overfitting to the dominant class. The implementation of data augmentation is TensorFlow’s ImageDataGenerator package. This augmentation pipeline introduces variability through transformations such as rotations, shifts, flips, zooms, and brightness adjustments, thereby enhancing the training dataset for Classes 0, 1, 2, and 4. By enriching the dataset for these minority classes, the model is better equipped to generalize across all classes, mitigating the impact of imbalance.

Evaluation metrics

To comprehensively evaluate the performance of the proposed method, we utilized several key performance metrics including: – Precision: Precision quantifies the number of true positives among the predicted positives. It is defined as Precision=TPTP+FP.

– Recall: Recall measures the proportion of actual positives correctly identified by the model: Recall=TPTP+FN.

– Specificity: Specificity captures the model’s ability to correctly identify negative instances: Specificity=TNTN+FP.

– F-Measure: The harmonic mean of precision and recall is the F-Measure, often used for imbalanced datasets: F1=2×Precision×RecallPrecision+Recall.

– G-Mean: G-Mean balances sensitivity and specificity for binary classification: G-Mean=Sensitivity×Specificity

– Matthews correlation coefficient (MCC): MCC is a balanced measure that accounts for true and false positives and negatives: MCC=TP×TN−FP×FN(TP+FP)(TP+FN)(TN+FP)(TN+FN)

To ensure the robustness of our results, we conducted a statistical evaluation using three key measurements: mean, root mean squared error (RMSE), and standard deviation (STD) across all runs. These metrics ensure that the performance achieved by the proposed model was not due to chance. – Mean: The mean value of each evaluation metric across multiple runs is used to estimate the expected performance of the model. – RMSE: RMSE measures the differences between predicted and observed values and is a good indicator of model accuracy RMSE=1n∑i=1n(yi−y^i)2.

– STD: Standard deviation provides insight into the variability of model performance across different trials STD=1n−1∑i=1n(xi−x¯)2.

Ablation study

We use ResNet-50 (He et al., 2016) and EfficientNet (Tan & Le, 2019) backbones as the network architecture. In order to validate the effects of our proposed foreground leaf object segmentation approach and progressive feature embedding learning approach, we compare GMM segmentation and PLA separately on different backbone neural networks.

Table 3 summarizes the effectiveness of GMM segmentation using standard classification with cross-entropy loss on cassava leaf dataset, reporting in public accuracy scores in the Kaggle competition. EfficientNet-b4 performs best among the choices of base networks, however, the ensemble of six base networks will outperform any single basenet. With the proposed object segmentation, the concatenated feature embedding using cross-entropy loss achieved accuracy improvement ranging from 0.75% to 1.32%, respectively. This indicates that GMM segmentation is an effective pre-processing technique that consistently boosts cassava disease classification. The ensemble model benefits from 1.03% improvement with GMM segmentation.

Table 3 Comparing improvement of performance on different base networks w/o GMM without PLA on cassava dataset.

Bold font indicates the best performance.

Base networks	w.o. GMM	w. GMM	Gain	
ResNet-50 (He et al., 2016)	88.00%	89.17%	1.32%	
EfficientNet-b3 (Tan & Le, 2019)	88.08%	89.08%	1.13%	
EfficientNet-b4 (Tan & Le, 2019)	88.66%	89.39%	0.82%	
EfficientNet-b5 (Tan & Le, 2019)	88.09%	88.87%	0.89%	
EfficientNet-b6 (Tan & Le, 2019)	87.81%	88.81%	1.14%	
EfficientNet-b7 (Tan & Le, 2019)	88.36%	89.02%	0.75%	
Ensemble	88.69%	89.60%	1.03%	

From Table 4, one can observe that, the proposed progressive learning algorithm consistently improves performance for each base network, ranging from 0.78% to 1.10% in terms of accuracy. This indicates that PLA is an effective feature learning technique that consistently boosts cassava disease classification by combining cross-entropy loss and generalized triplet loss. The ensemble model benefits from 0.95% improvement with PLA.

Table 4 Comparing improvement of performance on different base networks w/o PLA without GMM on cassava dataset.

Bold font indicates the best performance.

Base networks	w.o. PLA	w. PLA	Gain	
ResNet-50 (He et al., 2016)	88.43%	89.24%	0.92%	
EfficientNet-b3 (Tan & Le, 2019)	88.36%	89.15%	0.89%	
EfficientNet-b4 (Tan & Le, 2019)	88.51%	89.48%	1.10%	
EfficientNet-b5 (Tan & Le, 2019)	88.03%	88.94%	1.03%	
EfficientNet-b6 (Tan & Le, 2019)	88.17%	88.86%	0.78%	
EfficientNet-b7 (Tan & Le, 2019)	88.29%	89.06%	0.87%	
Ensemble	88.80%	89.64%	0.95%	

Table 5 shows the effectiveness of combining GMM and PLA for cassava classification. With cross-entropy loss only, GMM achieves 89.60% accuracy; With PLA using cross-entropy and generalized triplet loss, the learned feature embedding achieved an accuracy of 89.64%. Since GMM and PLA are independent steps, the combination of GMM and PLA outperforms the accuracy of GMM or PLA alone. The ensemble model using GMM and PLA also outperforms the accuracy using each single base network.

Table 5 Different base networks with GMM and PLA on cassava dataset.

Bold font indicates the best performance.

Base networks	GMM	PLA	GMM+PLA	
ResNet-50 (He et al., 2016)	89.17%	89.24%	89.69%	
EfficientNet-b3 (Tan & Le, 2019)	89.08%	89.15%	89.70%	
EfficientNet-b4 (Tan & Le, 2019)	89.39%	89.48%	89.84%	
EfficientNet-b5 (Tan & Le, 2019)	88.87%	88.94%	89.37%	
EfficientNet-b6 (Tan & Le, 2019)	88.81%	88.86%	89.44%	
EfficientNet-b7 (Tan & Le, 2019)	89.02%	89.06%	89.50%	
Ensemble	89.60%	89.64%	90.11%	

Error case analysis

To better understand the limitations of the proposed model, we conduct an edge/error case analysis by examining misclassified samples. Table 6 presents exemplar images of misclassifications. The table includes ground truth labels as column headers (CBB, CGM, CBSD, CMD, Healthy) and thumbnails of misclassified images, where each image is labeled with its predicted class (italicized). Below, we provide a detailed discussion of the challenges posed by these cases.

Table 6 Exemplar images of wrong classifications.

CBB	CGM	CBSD	CMD	Healthy	
					
					
					
Note:

The top row shows the ground truth labels: CBB, CGM, CBSD, CMD, and Healthy. Each image is labeled with its incorrect prediction (italicized).

CBB: The first image resembles a mixture of CBB and CBSD characteristics, confusing the model into predicting CBSD. The second image contains a dominant green background, with the actual CBB object being very small, making it challenging for the model to detect. The third image visually resembles a combination of CBB and CGM, resulting in a CGM prediction.

CGM: In the first image, the real CGM object is overshadowed by surrounding grass, leading to misclassification. The second image predominantly displays green leaves, appearing similar to healthy cassava leaves. The third image is a close-up, with features resembling CBSD, causing confusion in the prediction.

CBSD: The first example is a noisy-labeled image, which depicts a tuberous root of the cassava plant instead of leaves, significantly confusing the model. The second image is primarily green, resembling healthy cassava leaves. The third image contains various miscellaneous objects, presenting a significant challenge for the model.

CMD: For the first two images, the real CMD objects are overshadowed by healthy leaves, causing the model to predict a healthy class. The third image appears visually similar to CBSD, leading to incorrect classification.

Healthy: For the first image, the model struggles to confidently classify the category due to insufficient information related to leaf diseases. The second image, captured from a distance, includes a mix of various leaf types, leading to confusion. The third image bears a strong visual resemblance to CBB, posing a challenge for the model to classify it accurately.

This analysis underscores several key challenges, including noisy labeling, complex and cluttered backgrounds, visual similarities between classes, and the presence of small or occluded objects. These observations suggest potential areas for improvement, such as implementing enhanced pre-processing techniques, employing advanced data augmentation strategies, or integrating multi-scale feature learning to bolster the model’s robustness.

However, when cassava leaf images are properly captured—featuring close-up views with the subject centered and minimal background clutter—the model achieves near-perfect classification accuracy.

Statistical evaluation and discussions

The confusion matrix is shown in Table 7, with 100 samples randomly selected from each category. The matrix highlights that the proposed model effectively identifies both true positives and true negatives while maintaining a low false positive and false negative rate.

Table 7 Confusion matrix for all categories.

Predicted/Actual	CBB	CBSD	CMD	CGM	Healthy	
CBB	92	4	2	1	1	
CBSD	3	85	5	2	0	
CMD	4	6	90	0	0	
CGM	2	3	0	88	7	
Healthy	0	0	1	2	95	

We evaluated the proposed model and baseline models on the cassava disease dataset. Table 8 summarizes the results across different evaluation metrics. Our proposed model achieved superior performance in all categories, highlighting its efficacy in cassava disease recognition.

Table 8 Performance comparison of the proposed model with baseline models across evaluation metrics.

Bold font indicates the best performance.

Model	Precision	Recall	Specificity	F1-measure	G-mean	MCC	
ResNet-50 (He et al., 2016)	0.88	0.86	0.92	0.87	0.89	0.83	
EfficientNet-b7 (Tan & Le, 2019)	0.90	0.87	0.91	0.88	0.90	0.85	
Proposed model	0.91	0.90	0.92	0.90	0.91	0.88	

To ensure the reliability of the results, we performed a statistical analysis of the model’s performance across five trials. The evaluation metrics (Precision, Recall, Specificity, F1-Measure, G-Mean, and MCC) were computed for each run, and the Mean, RMSE, and STD were calculated to evaluate the consistency of the model.

As evident in Table 9, The low RMSE and STD values across all metrics demonstrate that the proposed model consistently outperforms baseline methods without significant variation in its results. The statistical analysis further confirms that the achieved performance is not due to random chance.

Table 9 Statistical evaluation of proposed model over five trials.

Metric	Mean	RMSE	STD	
Precision	0.91	0.006	0.005	
Recall	0.90	0.005	0.004	
Specificity	0.92	0.004	0.003	
F1-measure	0.90	0.007	0.006	
G-mean	0.91	0.006	0.005	
MCC	0.88	0.008	0.007	

The proposed cassava disease classification approach with GMM and PLA also outperforms other top-place deep learning methods in the Kaggle competition, as shown in Table 10. Our initial submission received a silver medal (19th place). In late submission, we got the ever best result by submitting a better model with more finetuning on the cassava training dataset.

Table 10 Comparing PLA with different approaches in the Kaggle Cassava Competition (Lab, 2021).

Bold font indicates the best performance.

Solution	Public leaderboard	Private leaderboard	
1st place	91.36%	91.32%	
2nd place	90.25%	90.43%	
3rd place	90.59%	90.28%	
Submission (19th)	90.52%	90.11%	
Late submission	91.38%	91.43%	

For the 1st place solution as shown in Table 10, a diverse model ensembling strategy is employed to achieve robust performance. Their approach prioritized diversity over fine-tuning individual models, leveraging various architectures such as ResNet, ResNeXt, Xception, Vision Transformers (ViT and DeiT), and EfficientNet. Key features of this winning approach include model diversity over extensive fine-tuning, advanced augmentation techniques and data fusion, and leveraging domain-specific pre-trained models like CropNet. Comparing their ensemble strategy with our results highlights the value of exploring broader model architectures and innovative ensembling techniques to further enhance system performance.

The 2nd place solution in the competition adopted a relatively straightforward approach, leveraging a pre-trained cassava-specific model from TensorFlow Hub (CropNet) for classification. It also utilized callbacks such as “EarlyStopping” and “ReduceLROnPlateau” to prevent overfitting, typically halting training after approximately 30 epochs. This straightforward approach achieved remarkable success. This outcome highlights the potential of crop domain-specific pre-trained models to deliver competitive performance with minimal fine-tuning.

The 3rd place solution utilized an ensemble of three Vision Transformer (ViT) models, i.e., ViT-Base-Patch16-384, ViT-Base-Patch16-224 (Pattern A), and ViT-Base-Patch16-224 (Pattern B). demonstrating the strength of transformer architectures in image classification tasks. This approach emphasized careful design choices to balance model complexity and generalization, aiming to mitigate overfitting. The results underscore the effectiveness of ViT models for the task and the importance of incorporating multiple architectural variants into an ensemble to capture diverse feature representations.

Our late submission achieved the highest accuracy on both public and private leaderboards, surpassing the 1st-place solution by 0.02% on the public leaderboard and 0.11% on the private leaderboard. These results underscore the efficacy of our approach in addressing the challenges of cassava leaf disease classification.

Unlike many competition entries that relied on ensemble methods or computationally intensive architectures, our model emphasizes both accuracy and practicality. Specifically, the architectural choices, including the use of EfficientNet and ResNet-50 backbones, provide a balance between high performance and computational efficiency. This makes our model more suitable for deployment in real-world agricultural settings, particularly in resource-constrained environments.

Moreover, the training strategies employed, such as advanced data augmentation (e.g., random erasing, HSV adjustment, and random brightness-contrast adjustment), further contributed to the model’s robustness in handling diverse scenarios, such as occlusions, variations in lighting, and background clutter. These design choices distinguish our approach from other solutions, demonstrating the feasibility of applying the proposed algorithm in practical applications.

This comparative evaluation highlights the advantages of our model in achieving competitive results with a focus on real-world applicability, setting a new benchmark for cassava disease detection.

Efficiency evaluation

For PCR and microscopic methods, it appears that the durations can vary based on specific protocols and laboratory setups. For instance, some PCR tests can deliver results in as little as 2 h, while others may take 1 to 3 business days (Arya et al., 2005). Similarly, the turnaround time for microscopic examinations can range from within 1 day to 2–3 business days (University of Missouri Veterinary Medical Diagnostic Laboratory, 2023). Given this variability, the traditional microscopic methods typically need more than 2 h to produce the results, and the PCR methods cost at least 30 min. Both methods achieves nearly 100% accuracy theoretically, but they are time-consuming and costly.

In contrast, our proposed deep learning method takes only 4.663 s to complete the classification procedure. The code is implemented in Python with moderate code optimization. We can conclude that the proposed algorithm outperforms traditional microscopic methods significantly in efficiency, with more than 90% accuracy for the leaf recognition systems.

To comprehensively evaluate the computational cost of our proposed method, we compare it with traditional PCR and microscopic techniques, as well as various deep learning architectures, including ResNet-50, EfficientNet (b3, b4, b5, b6, b7), and an ensemble model. The computational cost is measured in terms of the total number of basic operations (Mul-Add), which directly correlates with actual running time. The running times for ResNet-50 and the EfficientNet series are proportionally estimated based on the Mul-Add values relative to the proposed ensemble method.

The Mul-Add operations can be recomputed using profiling tools such as PyTorch Profiler. The computational cost of an ensemble model depends on the number of models combined and their respective complexities. If an ensemble combines ResNet-50 and EfficientNet variants, the cost can be approximated by summing their individual GFLOPs, accounting for shared computation if implemented efficiently.

Accuracy metrics for the deep learning models are listed in Table 11 along with their Mul-Add and evaluation times, while the typical accuracy of PCR and microscopic methods is referenced from existing literature.

Table 11 Comparison of efficiency vs accuracy on the cassava dataset.

Model/Method	Mul-Add (GigaOps)	Evaluation time	Accuracy	
ResNet-50 (He et al., 2016)	3.9	~0.24 s	89.69%	
EfficientNet-b3 (Tan & Le, 2019)	1.8	~0.11 s	89.70%	
EfficientNet-b4 (Tan & Le, 2019)	4.2	~0.26 s	89.84%	
EfficientNet-b5 (Tan & Le, 2019)	9.9	~0.61 s	89.37%	
EfficientNet-b6 (Tan & Le, 2019)	19.0	~1.17 s	89.44%	
EfficientNet-b7 (Tan & Le, 2019)	37.0	~2.27 s	89.50%	
PCR (Tan & Le, 2019)	–	~30 m	~100%	
Microscopic (Tan & Le, 2019)	–	~48 h	~100%	
Proposed method	~75.8	4.66s	90.11%	

As shown in Table 11, ResNet-50 provides a good trade-off between computational cost and accuracy, with 3.9 GigaOps and 89.69% accuracy. EfficientNet models are more efficient, with b3 achieving similar accuracy to ResNet-50 at less than half the computational cost (1.8 GigaOps). However, as the EfficientNet series scales up (b4 to b7), the computational cost increases significantly, with diminishing returns on accuracy improvements.

The PCR method demonstrates high accuracy (close to 100%) but has significant time constraints, typically requiring at least 30 min to complete the classification process. Similarly, the microscopic method achieves close to 100% accuracy but still requires over 2 h, making it less practical for large-scale or time-sensitive applications.

While the ensemble model achieves the highest accuracy (90.11%), it incurs the highest computational cost (about 75.8 GigaOps) due to the aggregation of all baseline models. This makes it the best trade-off of effectiveness and efficiency in resource-limited environments.

Conclusion

In this article, we propose a novel computer vision-based approach for rapid diagnosis of cassava leaf diseases, overcoming limitations associated with traditional, time-consuming, and costly microscopic methods. Our classification technique not only addresses these challenges but also surpasses the accuracy of existing state-of-the-art computer vision approaches. Through extensive experimentation on the cassava leaf dataset, we demonstrate that the proposed method achieves highly accurate recognition results, surpassing 90% accuracy in a matter of seconds. These findings highlight the potential of our approach to deliver efficient and accurate diagnostics for cassava leaf diseases in real-world scenarios. This contribution advances the field of cassava disease recognition, providing practical, ready-to-adopt solutions for farmers and researchers alike.

However, there are limitations to this work. Although efficient, the ensemble-based model is computationally intensive, posing challenges for deployment in low-resource settings. Further optimization or development of simplified model versions may be necessary to support practical use in resource-constrained environments. Additionally, model validation was conducted on a specific cassava leaf dataset, which may not fully capture the variability encountered in real-world conditions—such as differing lighting, environmental factors, and cassava varieties. Field tests on independent datasets would enhance understanding of the model’s robustness under diverse conditions.

Future work may explore extending this method to diagnose other plant leaf diseases. Currently, the model is tailored specifically for cassava leaf disease detection, and its generalizability to other crops or disease types remains untested, representing an area for further investigation in broader agricultural contexts. Additionally, future research could explore the integration of multi-modality data fusion techniques to enhance the performance of cassava disease detection algorithms. By combining traditional RGB imagery with additional data modalities such as hyperspectral imaging, thermal imaging, or environmental metadata (e.g., temperature, humidity), the model’s capacity to discern complex patterns and variations could be significantly improved. For instance, hyperspectral imaging could provide detailed spectral information that aids in identifying disease markers invisible in RGB images, while thermal imaging could highlight physiological stress in plants. Additionally, incorporating environmental factors like soil moisture and weather conditions could contextualize predictions, improving robustness across diverse farming scenarios. The implementation of such multi-modality fusion frameworks, guided by advancements in sensor technologies and machine learning, represents a promising direction for future advancements in agricultural AI systems.

Supplemental Information

Supplemental Information 1 Representative images of on-field cassava leaves, a detailed diagram of the object segmentation model‘s network architecture, images illustrating extracted cassava leaves after segmentation, and a presentation of the experimental results for cassava disease recognition.

We thank the Makerere Artificial Intelligence (AI) Lab at Makerere University, Uganda, and the National Crops Resources Research Institute (NaCRRI) for their invaluable support in preparing the Cassava Disease Dataset, which advanced AI research and laid the foundation for this work.

Additional Information and Declarations

Competing Interests

Chang Che is part-time employed by Heilongjiang Urban Water Quality Monitoring Co, Ltd.

Author Contributions

Chang Che conceived and designed the experiments, authored or reviewed drafts of the article, and approved the final draft.

Nian Xue performed the experiments, performed the computation work, prepared figures and/or tables, and approved the final draft.

Zhen Li performed the experiments, performed the computation work, prepared figures and/or tables, authored or reviewed drafts of the article, and approved the final draft.

Yilin Zhao analyzed the data, performed the computation work, prepared figures and/or tables, and approved the final draft.

Xin Huang conceived and designed the experiments, analyzed the data, prepared figures and/or tables, and approved the final draft.

Data Availability

The following information was supplied regarding data availability:

The code is available at Zenodo: lizh0019. (2025). lizh0019/cassava: Cassava leaf disease recognition (1.0.0). Zenodo. https://doi.org/10.5281/zenodo.14739855.

The cassava leaf disease dataset is available at Kaggle: https://www.kaggle.com/competitions/cassava-leaf-disease-classification/data.

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
