# Peer review of "Automatic cassava disease recognition using object segmentation and progressive learning"

_PeerJ Computer Science, doi:10.7717/peerj-cs.2721_

## Round 0.1 · original submission · Minor Revisions

Dear Authors,

Your paper has been revised. Minor revisions are needed before it is considered for publication in PEERJ Computer Science. More precisely, the following issues need to be addressed:

1) The authors should better justify using ResNet-50 and EfficientNet backbones as the network architecture.
2) The authors should demonstrate whether the proposed algorithm is resistant to extreme environmental degradation, such as low light, noise, haze, etc.
3) The authors should add a multi-modality data fusion method for future research.

Reviewer 1 ·

Basic reporting

The paper proposes a deep learning-based approach to classify cassava leaf diseases. It introduces a hybrid pipeline combining Gaussian Mixture Model-based segmentation, a self-supervised U-Net for refined segmentation, and a Progressive Learning Algorithm for robust feature embedding. The ensemble of EfficientNet models further enhances classification accuracy. The method achieves over 90% accuracy on the Cassava Leaf Disease Classification dataset, significantly outperforming other models in the Kaggle competition.

Experimental design

To me, several problems should be addressed.

(1) One of the most important evaluation metrics is the efficiency of the proposed method, which should be added to this paper.

(2) this paper's problem is highly relevant to a widely developed field, camouflaged object detection. Several cutting-edge strategies are recommended to be referred to (or at least discussed in the related works), for example, feature filtering (such as the wavelet decomposition in "Camouflaged object detection with feature decomposition and edge reconstruction"), feature aggregation (such as multi-scale feature grouping in "Weakly-supervised concealed object segmentation with sam-based pseudo labeling and multi-scale feature grouping"), and introducing auxiliary tasks (such as the introduction of the auxiliary edge reconstruction task in "Strategic preys make acute predators: Enhancing camouflaged object detectors by generating camouflaged objects").

(3) The reviewer wonders whether the proposed algorithm can resist extreme degradation, such as low light, noise, haze, etc. One paper can be referred to: Reti-Diff: Illumination Degradation Image Restoration with Retinex-based Latent Diffusion Model.

(4) Multi-modality data fusion is considered a promising way to improve the robustness of the proposed method. The authors are suggested to add this as a future direction.

Validity of the findings

See the detailed requirements in "Experimental design".

·

Basic reporting

While this paper presents a promising approach to cassava disease detection, several key aspects of basic reporting require improvement to meet scientific standards. The methodology section needs more comprehensive details about hyperparameter settings, model architectures, and training protocols that would enable reproducibility, especially regarding the implementation specifics of ResNet50 and EfficientNet models. The handling of the significant class imbalance shown in Table 1 is inadequately addressed, with insufficient explanation of any mitigation strategies employed during training. The evaluation methodology lacks rigor, particularly in detailing the train/validation/test split ratios and any cross-validation procedures used. Additionally, while accuracy metrics are presented, the paper would benefit from statistical significance testing and more thorough error analysis. The comparative analysis with baseline methods, though showing improvements, needs deeper examination and contextualization within the current state-of-the-art. These enhancements would significantly strengthen the scientific reporting and enable better reproducibility of the research.

Experimental design

The experimental design of this paper exhibits several significant limitations that should be addressed. While the authors present a novel approach combining object segmentation and progressive learning, the experimental validation lacks the necessary rigor and comprehensiveness expected in a scientific publication. The authors do not provide adequate justification for their choice of hyperparameters, network architectures, or loss function combinations, and there is no ablation study to demonstrate the individual contribution of each component in their proposed system. The comparison with baseline methods is limited and does not include recent state-of-the-art approaches in plant disease detection. The paper also lacks cross-dataset validation to demonstrate the generalizability of their method, and there is insufficient analysis of the model's performance under different real-world conditions such as varying lighting, image quality, or disease severity levels. The evaluation metrics, while including accuracy, precision, and recall, would benefit from additional metrics such as inference time and computational requirements that are crucial for practical deployment. Furthermore, the statistical significance of the reported improvements is not thoroughly analyzed, and there is no discussion of the limitations or potential failure cases of the proposed method.

Validity of the findings

The validity of the findings in this paper raises several concerns that need to be addressed to meet scientific standards. While the reported accuracy of 91.43% is impressive, the lack of comprehensive error analysis and statistical validation weakens the reliability of these results. The authors do not adequately address potential biases in their dataset, particularly given the significant class imbalance, and there is insufficient discussion of how this might affect real-world performance. The paper's claims about the superiority of their ensemble approach would be more convincing with rigorous statistical testing to demonstrate significance. The experimental validation is primarily conducted on a single dataset under controlled conditions, raising questions about generalizability to real-world scenarios with varying environmental conditions, lighting, and disease manifestations. Additionally, the authors should include a more detailed analysis of failure cases and edge scenarios, as well as provide confidence intervals for their reported metrics. The improvement over baseline methods, while numerically demonstrated, lacks sufficient explanation of why and how their approach achieves better results. Furthermore, the computational requirements and processing time for the ensemble model are not adequately discussed, making it difficult to assess the practical feasibility of the proposed solution for real-world agricultural applications.

Additional comments

No comments

Reviewer 3 ·

Basic reporting

The paper has a good coherent, well organized in term of the structure, and has good language.
Abstract - Abstract is short but contains all the required elements.
Introduction – Good motivation and the direction of the study is provided. Authors also have provided clear contributions of their study.
Literature Review – Additional references can be provided to provide clear gaps of the current works. Good to have additional comparison between the current studies.

Experimental design

- Methods are described in sufficient and detail.
- Authors suggest that “One of the key contributions of our paper is the development of the first deep learning approach for the recognition of common cassava leaf diseases.” – Does the paper used to enhance current approach or develop a new novel approach? Need to clarify.

Validity of the findings

Line 307 - We use ResNet-50(He et al., 2016) and EfficientNet (Tan and Le, 2019) backbones as the network architecture. Why? Please justify.
Further discussion needed to make a comparative evaluation between the proposed model and current models.

---

## Round 0.2 · accepted · Accept

Dear Authors,

Your paper has been revised. It has been accepted for publication in PEERJ Computer Science. Thank you for your fine contribution.

Reviewer 1 ·

Basic reporting

This paper has promising novelty with clear motivations. Additionally, the whole content is fluent and well-written. The results demonstrate the superiority of the proposed method.

Experimental design

The experiment is sufficient with promising performance. The implementation details are clear.

Validity of the findings

The finding of this paper is promising. Therefore, I recommend accepting this paper!

·

Basic reporting

No comment

Experimental design

No comment

Validity of the findings

No comment

Additional comments

The paper can be accepted in current form